# Professional Nurses’ Experiences Regarding Antiretroviral Adherence by Incarcerated Individuals Living with HIV and AIDS in Correctional Services

**DOI:** 10.3390/ijerph22121772

**Published:** 2025-11-21

**Authors:** Ntavhanyeni Mapholisa, Mankuku Mary Madumo, Tshimangadzo Selina Mudau, Nkhensani Florence Mabunda

**Affiliations:** 1Nursing Science Department, School of Health Care Sciences, Sefako Makgatho Health Sciences University, Molotlegi St., Ga-Rankuwa 0208, South Africa; ntavhanyenimapholisa@gmail.com (N.M.); marymadumo36@gmail.com (M.M.M.); 2Nursing Department, School of Nursing, University of KwaZulu-Natal, 238 Mazisi Kunene Road, Durban 4041, South Africa; selimgc4@gmail.com; 3Department of Development Studies, College of Human Sciences, University of South Africa, Preller St., Pretoria 0002, South Africa

**Keywords:** adherence, ART, professional nurses, experiences, incarcerated individuals, correctional facilities

## Abstract

Background: Antiretroviral adherence for incarcerated individuals living with HIV/AIDS in correctional service facilities remains a challenge. This study examined the experiences of professional nurses related to this issue in Limpopo Province, South Africa. Method: A qualitative, exploratory, descriptive, and contextual approach was employed in the Correctional Services Department facilities within the Vhembe District. The professional nurses were purportedly sampled. Data were collected through semi-structured telephone interviews and analysed thematically. Results: Three themes emerged: (1) professional nurses’ experiences with incarcerated individuals from foreign countries; (2) manipulative behaviours; and (3) misuse of antiretroviral therapy (ART) medication by incarcerated individuals. Conclusion: For public policy, the findings require the development of standardised guidelines for the management of foreign national incarcerated individuals and the implementation of anti-diversion strategies to prevent misuse of medications. For nursing practice, the results emphasise the importance of specialised training programmes that equip nurses to manage manipulative behaviours, enhanced supervision systems addressing moral distress, and structured adherence monitoring, including direct observed therapy for high-risk incarcerated individuals. These evidence-based interventions are essential to improve the outcomes of ART adherence, reduce treatment failure and drug resistance, and decrease HIV-related mortality in correctional settings while protecting general public health.

## 1. Introduction

The HIV/AIDS pandemic remains a significant global public health challenge, particularly in sub-Saharan Africa, where 39.9 million [36.1–44.6 million] people lived with HIV in 2023 [1]. Despite progress in expanding HIV testing, prevention, and antiretroviral treatment (ART) worldwide, HIV-related deaths remain high [2]. Sustained adherence to ART is crucial for improving health outcomes. The Joint United Nations Programme on HIV/AIDS has established 95-95-95 targets by 2030: 95% of people living with HIV diagnosed, 95% of those diagnosed on ART, and 95% of those on ART virally suppressed.

South Africa has the highest HIV prevalence and incarceration rate in sub-Saharan Africa and the largest population of people living with HIV globally [3,4,5,6]. HIV prevalence in South African correctional facilities ranges between 9% and 41% [7], with incarcerated individuals experiencing higher infection rates than the general population [8]. Optimal adherence to ART is essential for maintaining viral suppression among incarcerated individuals [9], yet correctional facilities face unique barriers to treatment adherence [3]. Treatment adherence of 95% or higher is necessary to achieve viral suppression, prevent drug resistance, and avoid opportunistic infections [3,4,10].

ART adherence refers to the extent to which a person’s medication intake behaviour corresponds to the recommendations of healthcare professionals and is critical to achieve viral suppression, maintain immune function, and prevent drug resistance [10]. Multiple factors influence adherence in correctional settings, including stigma and discrimination, lack of privacy, poor healthcare access, food insecurity, low levels of education, substance abuse, lack of family support, overcrowding, inadequate training of staff, and fear of disclosure of HIV status [11,12]. Professional nurses who manage HIV-positive incarcerated individuals face challenges, including non-compliance with the offender, inadequate infrastructure, and inadequate staffing [12,13,14]. Despite their critical role in the management of ART adherence, the literature on nurses’ lived experiences in this context is limited, particularly in South Africa.

This study aimed to explore and describe the experiences of professional nurses’ in regard to antiretroviral adherence by incarcerated individuals living with HIV/AIDS in correctional service facilities. Understanding these challenges is essential to improve care and reduce HIV/AIDS-related mortality in correctional healthcare settings.

## 2. Materials and Methods

### 2.1. Research Design

The research study used a qualitative, exploratory, descriptive and contextual design to obtain an in-depth understanding of nurses’ experiences with respect to adherence to ARV among incarcerated individuals in correctional service facilities [15,16]. Semi-structured individual telephone interviews were conducted with 22 professional nurses to encourage them to tell their experiences of adherence to ART for incarcerated individuals living with HIV in the settings of the correctional facilities. They answered the research question, “What are the ARV adherence experiences of incarcerated individuals with HIV/AIDS in correctional service facilities?”.

### 2.2. Study Setting

The research study was conducted in the Department of Correctional Services (DCS) in Limpopo Province. The two correctional service Management Areas, A and B, with the highest number of incarcerated individuals among all provincial correctional facilities, were sampled for the study. The Correctional Services Department facilities are located in the Vhembe District, the northern side of the border in Limpopo, South Africa. The Management Area A consisted of four prisons or correctional centres, each with a PHC clinic on its premises. Both sentenced and awaiting trial incarcerated individuals were incarcerated in these correctional facilities. Management Area B is a semi-private correctional centre that houses male inmates who are sentenced to 12–25 years. In February 2020, the total number of inmates in ARVs was 626. The total population of all individuals incarcerated in both prisons was 3024 at the end of February 2020. An estimated 1000 inmates are seen at the PHC daily in Management Area B.

### 2.3. Population and Sampling

The study population and sampling consisted of a total of 19 professional nurses, including both males and females. Non-probability purpose-sampling was used to select professional nurses working in correctional services facilities, specifically Management Areas A and B. PARTicipants were sampled with the author’s assumption that they have more experience with ART adherence for incarcerated individuals living with HIV in a prison setting. Inclusion criteria included being a registered professional nurse, having at least six months of experience in a correctional healthcare setting, and having direct involvement in the management of antiretroviral therapy for incarcerated individuals. Nurse-Initiated on the Management of Antiretroviral Therapy (NIMART) trained and willing to participate in the study telephone interview process. All other professional nurses who did not meet the inclusion criteria were excluded.

### 2.4. Data Collection

Data were collected by the researcher at the DCS Management Area A and B premises using semi-structured interviews. This method was selected for its flexibility and ability to elicit in-depth, narrative-rich responses while allowing professional nurses to explore the ARV adherence experience during interviews. In-depth Individual telephone interviews with professional nurses working in correctional healthcare facilities were conducted to explore and describe lived ARV adherence experiences for incarcerated individuals with HIV/AIDS. The professional nurses interested in participating in the research study were recruited with the approval of the National Department of Health, the Area Commissioner, and management representatives of the facilities in Management Area A and B, as well as the Head of Correctional Centres (HCC), Nursing service managers and Operational Nursing managers in correctional service facilities.

The research study, the purpose of the study, and the significance of the study were introduced by managers during managerial meetings and the’ morning parades of officials before the actual recruitment of study participants. After the research study was introduced, the researcher met with participants in all correctional centres for recruitment. Participants were informed of the purpose of the study and gave their written informed consent before data collection. In-depth telephone interviews were scheduled at convenient times agreed upon by the researcher and the professional nurses, all of whom spoke English. Data were collected from May 2021 to September 2021. All interviews were recorded with the’ consent of the participants. Four females were interviewed for the pilot study in the two correctional service Management areas and then adjustments were made. The interviews lasted between 45 and 60 min, depending on the depth of the discussion. Orientated by data saturation, data collection continued until saturation was reached in 19 interviews.

To direct the participant during the telephone interview, the researcher developed an interview guide with semi-structured questions [17,18] with a central question: “What are the experiences of professional nurses with respect to ARV adherence by incarcerated individuals living with HIV/AIDS in the correctional service?” Probes and follow-up questions were asked to encourage participants to elaborate on their responses and explore their ARV adherence experiences, as indicated below.

Could you elaborate on your experience in organizing and managing incarcerated individuals during their planned ART appointments?Could you elaborate on your experiences regarding adherence during ART refill and consultation sessions with individuals living with HIV/AIDS within this correctional facility?What are the common challenges you have encountered that affect incarcerated individuals’ adherence to ART?Are there any additional insights or observations you would like to contribute concerning your experiences with antiretroviral therapy adherence among the incarcerated individuals?

### 2.5. Data Analysis

The data analysis process for this study followed Tesch’s eight-step qualitative data analysis method [19,20]. These steps included (1) getting a sense of the whole; (2) selecting a single interview for initial analysis; (3) listing all topics; (4) clustering similar topics into themes; (5) coding the data; (6) categorising and naming themes; (7) organising themes and interrelationships; and (8) recoding. The first author read all of the transcripts repeatedly to gain an understanding of the data and jotted down the ideas or thoughts as they came to mind. Thematic analysis was conducted. Similar topics were clustered and coded. The coded data were then organised into themes and categories. The data were then compared for similarities and differences. The researcher and independent coder concluded the findings. No analytical software was used.

### 2.6. Ethical Considerations

Ethical clearance for this study was obtained from the Sefako Makgatho Health Science Research Ethics Committee (Reference no.: SMURECIH/226/2020: PG). Subsequently, a letter requesting permission to conduct the study was sent to the National Department of Correctional Services Research institution. The study was conducted in accordance with established ethical principles, including respect for persons, beneficence, non-maleficence, and justice. Participants were respected and informed about their right to participate in the study voluntarily, without coercion. Participants were able to choose a date and time convenient for conducting telephone interviews. Participants were informed that they could withdraw from participation at any time without punishment. To ensure justice, participants were treated equally without bias. Codes were assigned to all participants to protect their identity and participation in the study. Informed consent was obtained from each participant in writing before participating in the study. Data were only shared with the authors and the co-coder.

This study used five criteria proposed by Lincoln and Guba [21], as referenced by Flanagan and Beck [22], to establish the trustworthiness of the study: credibility, dependability, confirmability, and transferability. Credibility was established through extended telephone contact with participants until data saturation was achieved, as well as the use of recorded audio interviews to accurately represent the experiences of professional nurses. The transcripts of the telephone interviews were distributed to the supervisor, co-supervisor and an independent coder to enhance triangulation and validate the study findings. The researcher maintained a reflexive journal to document personal reflections, emotional responses, and emergent insights throughout the data collection and analysis process, thus ensuring reflexivity and methodological rigour.

During telephone interviews, the researcher reiterated and summarised the’ statements of the participants to verify the accuracy and authenticity of the information provided, thus facilitating the checking of the members. Dependability was ensured through the provision of a detailed description of research methods and the maintenance of a comprehensive audit trail, which included field notes, audio recordings, and verbal transcripts of telephone interviews throughout the research process. Confirmability was maintained by employing an audio recorder to capture the’ voices of the participants and an audit trail composed of field notes, audio-recorded data, and analysed data, ensuring that the findings accurately reflect the narratives of the participants’ rather than the researcher’s personal assumptions. In addition, the study utilised an independent coder to validate the data for accuracy, relevance and meaning. Transferability was addressed by providing a detailed description of the research setting and sample, employing a purposeful method to select professional nurses knowledgeable about the adherence of incarcerated individuals ‘to ARV, thereby enhancing the applicability of the findings to other contexts.

In addition, the authors’ position significantly influences knowledge production [23]. In this study, the authors, as insiders, have the advantage of gaining more access to the authentic stories of the participants; however, there is a risk of interpretive bias due to shared experiences. In contrast, an outsider might overlook subtle dynamics. Both viewpoints influence both observation and interpretation. In the context of prisons, the power dynamics did not influence the participants, who gave cautious responses out of fear of consequences. In addition, the authors engaged in reflexivity, assessing how their own biases and backgrounds influenced the study’s design to ensure the ethically accurate representation of nurses’ voices. Positionality underscores that findings are collaboratively constructed, impacted by the participants’ experiences, institutional cultures, and the researchers’ perspectives, thereby enhancing the study’s credibility through transparency.

In addition, the study was guided by the Consolidated Criteria for Reporting Qualitative Research (COREQ), a 32-item checklist designed to ensure complete and transparent reporting of qualitative studies involving interviews and focus groups [24,25]. This framework was adhered to throughout the research process to ensure methodological quality in the study methods, the study setting, the collection, analysis and the findings. The detailed description of the researcher’s reflexivity, participant selection, interview procedures, and analytical rigour were clearly documented to enhance the credibility and dependability of the study.

## 3. Results

Table 1 displays the demographic characteristics of the participants. Three main themes emerged from the thematic analysis, presented in Table 2. Findings are interpreted through Leventhal’s Common-Sense Model (CSM), which examines how cognitive and emotional representations of illness influence health behaviours.

### 3.1. Theme One: Professional Nurses’ Experiences Regarding Incarcerated Individuals from Foreign Countries

Nurses identified challenges in managing ART adherence for foreign national incarcerated individuals, relating to a lack of policies and staff unpreparedness.

#### 3.1.1. Sub-Theme One: Lack of Policies Addressing the Needs of Foreign Nationals

Professional nurses expressed concerns about the absence of policies and guidelines for foreign incarcerated individuals, which limits consistent healthcare delivery. Lack of documentation complicated communication, treatment planning, and continuity of care. The DCS system lacks standardised guidelines for the management of foreign nationals, resulting in discontinuity of care.

‘Yooo! I have forgotten something (hitting the head with the palm of the right hand) about foreigners. Sometimes we admit incarcerated individuals from foreign countries, such as Zimbabwe and Mozambique, among others. During admission, when tested for Tuberculosis (TB), HIV and sexually transmitted infections (STIs), the offender will say: “I am HIV-positive and take ARTs in Zimbabwe”. When you request a transfer letter, they respond that they do not have it. How can you, as a nurse, trace the transfer letter from a foreign country?(Female, 52-year-old professional nurse)

‘The other challenge in adherence to ARV in this correctional centre is the foreign incarcerated individuals. Some of them will say they are HIV-positive in ARTs. As professional nurses, we cannot perform ART follow-up in Zimbabwe; therefore, we should reinitiate incarcerated individuals in ARVs. Foreigners are mostly sentenced for short periods. Next month, when the ARV due date for the ART refill is, the offender will waste time re-initiating ART.(Female, 44-year-old professional nurse)

This policy vacuum affects nurses’ control beliefs (perceived ability to manage adherence effectively) and their cognitive appraisal of professional capacity. Without institutional protocols, nurses cannot accurately assess a patient’s treatment history (illness identity), leading to emotional responses of frustration and feelings of professional inadequacy. This systemic barrier hinders the effective implementation of coping strategies, leading to inconsistent care delivery and potential treatment interruptions that increase the risk of resistance.

#### 3.1.2. Sub-Theme Two: Staff Unpreparedness

Staff unpreparedness hampered nurses’ ability to manage ARV adherence. Nurses expressed difficulty handling complex psychological and behavioural challenges, struggling with confidence in decision-making, and feeling overwhelmed by adherence demands.

“Then stay with your ARV tablets and drink them”. The offender will then walk away from the consulting room, walking faster and becoming angrier, leaving the official behind, and return to the holding cell, defaulting to ARVs. Adherence to ARV treatment is a challenge in the Department of Correctional Services (DCS) and correctional centres; sometimes you can call it a madhouse.(Female, 47-year-old professional nurse)

‘Incarcerated individuals came and threw their ART tuberculosis on the nurses’ table and said “I am no longer drinking ARVs until all my problems are solved”. As a professional nurse, I tried to do adherence counselling without effect, stopped ARTs for 3 days, returned, demanding ARTs, and said: “Now my problems are solved, I need my ARV medication”. Adherence to ART medications is very difficult in correctional services facilities.(Female, 54-year-old professional nurse)

Inadequate training undermines nurses’ control beliefs and self-efficacy in managing adherence challenges. Unpredictable offender behaviours challenge nurses’ cognitive schemas about patient-provider interactions, generating emotional responses of overwhelm and helplessness. This affects your coping strategies, shifting from proactive adherence support to reactive crisis management, ultimately compromising the consistency and quality of ART services.

### 3.2. Theme Two: Professional Nurses’ Experiences of Incarcerated Individuals’ Manipulative Behaviours

Professional nurses described manipulative behaviors that undermine ARV treatment adherence, including system manipulation, exploiting legal procedures, and misuse of grievance systems.

#### 3.2.1. Sub-Theme One: System Manipulation for Personal Gain

Nurses described manipulative behaviours that undermine adherence to ARV treatment, including system manipulation, exploiting legal procedures, and misuse of grievance systems.

‘One of the contributing factors to the non-adherence to ARV in the correctional centre is the therapeutic or special diet. The offender will explain to the professional nurse that” I need a special diet. If the professional nurse explains the therapeutic diet manual and the body mass Index rule in the DCS correctional centre, the offender will go back to their cell and stop taking ARVs because he wants food. Okay, the body mass index is a weight-to-height measurement used to assess whether the offender qualifies for therapeutic high-protein, high-kilojoule diets in DCS.(50-year-old professional nurse)

‘The other challenge we experienced in the PHC wellness clinic is that some incarcerated individuals overdose themselves with ARV treatment when they experience some life problems. Like last week, an offender living with HIV and AIDS overdosed and was admitted to our local referring hospital. Currently, she has been discharged and we have placed him on DOT every morning.(54-year-old professional nurse)

Incarcerated individuals’ illness representations are reshaped by the correctional context—HIV becomes a bargaining tool rather than primarily a health threat. Their consequence beliefs prioritise institutional benefits (food, privileges) over health outcomes. This reflects maladaptive coping strategies shaped by incarceration’s power dynamics, where health-seeking behaviour merges with survival tactics. For nurses, this creates emotional representations of mistrust and frustration, eroding the therapeutic relationship essential for adherence support.

#### 3.2.2. Sub-Theme Two: Exploiting Legal Procedures

Incarcerated individuals intentionally exploit legal procedures to interfere with ARV treatment, report nurses to Human Rights, threaten to default, and disrupt healthcare care routines for personal gain.

‘Last month, an offender reported that nurses refused to give him his ARV medication. He was taking his medication daily through DOT due to non-adherence and unrepressed viral load. He requested that he receive all his ARV medication. He stopped taking his ARVs, claiming that he cannot walk to the clinic daily. He threw his tablets onto the table next to me and then headed to the cell. The next few days, the offender wrote a letter to Human Rights complaining that nurses are not giving him his treatment. I was called to make the statement, but I explained the whole story to the Human Rights official, who dismissed the complaint and instructed that the offender continue with DOT treatment.’(52-year-old professional nurse)

‘The offender will then place the ARV tablet under the tongue and pretend as if he has swallowed the tablet. At a later stage, when you finish the consultation, when you leave the consultation room, you will find tablets on the floor thrown there by incarcerated individuals on their way out to their holding cells. By the time you collect their blood for HIV monitoring, your viral load of incarcerated individuals living with HIV and AIDS on DOT treatment is not suppressed. I do not think that the viral load can be unsuppressed when the offender is taking ARVs properly.’(39-year-old professional nurse)

CSM Interpretation: The correctional environment distorts standard perceptions of identity and control of the illness. Incarcerated individuals perceive control through institutional mechanisms rather than health behaviours. For nurses, these experiences generate emotional responses of moral distress and betrayal, leading to defensive coping strategies that may prioritise documentation and self-protection over therapeutic engagement, potentially creating a cycle of mistrust that further undermines adherence.

#### 3.2.3. Sub-Theme Three: Misuse of Grievance Systems

Incarcerated individuals exploit formal grievance procedures to manipulate medical decisions, file complaints to intimidate nurses, delay disciplinary action, or gain leverage. This misuse leads to unnecessary investigations and emotional strain on nurses.

‘I want to explain to you that incarcerated people living with HIV and AIDS did not receive ARV treatment because they wanted a certain therapeutic diet. Incarcerated individuals are very manipulative. They will tell the nurse that “I will not take my ARV medication until I receive a high-protein diet or other specific named therapeutic diet”. They complain that they saw other incarcerated individuals receiving a specific therapeutic diet without understanding why such a specific therapeutic diet was prescribed to such an offender. If the diet they are demanding is not prescribed, they stop taking ARV medication.’(42-year-old professional nurse)

‘The challenge of food plays a role in ARV adherence by incarcerated individuals living with HIV and AIDS. Most of the incarcerated individuals threatened and blackmailed the professional nurses that they would stop taking ARVs if they were not provided with enough food. They will always demand additional food to comply with ARV treatment. Some incarcerated individuals will demand to be served with a prescribed therapeutic diet, especially high-protein high-kilojoules to prevent ARV, no therapeutic diet, no ARV adherence from incarcerated people living with HIV and AIDS in the correctional centre PHC clinic.’(39-year-old professional nurse)

This represents a fundamental misalignment between the coherence of incarcerated individuals’ illness (understanding of the HIV/ART relationship) and behavioural priorities. The institutional context reshapes the consequences beliefs; immediate needs (food, safety) supersede long-term health outcomes. The grievance system, designed to protect rights, becomes a tool that distorts the therapeutic relationship, generating maladaptive coping behaviours in which adherence is weaponised for institutional negotiation rather than health preservation.

### 3.3. Theme Three: Professional Nurses’ Experiences of Misusing ARV Medication by Incarcerated Individuals

Professional Nurses identified ARV medication misuse as compromising treatment outcomes, including faking medical conditions, misusing prescribed medication, and demanding unnecessary transfers.

#### 3.3.1. Sub-Theme One: Faking a Medical Condition

Incarcerated individuals deliberately exaggerate medical symptoms to manipulate the healthcare system, defaulting on ARV medication, and fabricating conditions for medical attention.

‘Other incarcerated individuals will say that “I will not drink my ARV tablets because my painful bone in the lower leg is not yet fixed or operated on, so I will not take antiretroviral treatment.’(35-year-old professional nurse)

‘Some incarcerated individuals placed in Direct observation Treatment (DOT) did not receive their ARVs treatment, complaining that they cannot walk to correctional healthcare services daily to collect and drink their ARV since they experience side effects of medication.’(43-year-old professional nurse)

This behaviour demonstrates distorted causal beliefs about ART; incarcerated individuals attribute control over medication decisions to unrelated health conditions, reflecting poor coherence of the illness. It suggests that nurses must address fundamental gaps in incarcerated individuals’ illness representations, particularly in their cognitive understanding of ART’s independence from other health conditions and the consequences of treatment interruption.

#### 3.3.2. Sub-Theme Two: Misuse of Prescribed Medication

Incarcerated individuals sell ARV medication for drugs or exchange treatment for protection against bullying or sodomy by gangsters. This misuse undermines treatment adherence and leads to treatment and virological failure.

‘Okay, in the correctional centre where I work, only males are incarcerated. Incarcerated individuals engage in sexual relationships: sodomy. I am referring to a sexual relationship between men and men in the correctional centre. So, incarcerated individuals exchange their ARV with gang leaders to prevent them from being sodomised by bosses in the correctional centre holding cells, and this affects adherence to ARV medication. ARVs are used by these gangster leaders as drugs. Incarcerated individuals agreed to prevent themselves from being bullied by those in leadership positions.(48-year-old professional nurse)

‘Most incarcerated individuals stopped using ARV medication due to substance abuse. The challenge is in a two-way fold; some incarcerated individuals living with HIV and AIDS themselves use their own ARV treatment to make drugs out of it to smoke. The other way is that those incarcerated people who engage in gangsters will take the ARV for incarcerated people living with HIV and AIDS by force.’(39-year-old professional nurse)

Institutional violence and vulnerability fundamentally alter the consequences beliefs of incarcerated individuals’ consequence beliefs; immediate physical safety becomes more important than viral suppression. The emotional representation of fear (violence, sexual exploitation) completely overrides the cognitive understanding of the importance of ART. This demonstrates how the correctional context can profoundly reshape illness perceptions such that survival needs override health behaviours, necessitating interventions that address safety and protection in conjunction with adherence to medical care.

#### 3.3.3. Sub-Theme Three: Demanding Unnecessary Transfers

Incarcerated individuals deliberately request unnecessary medical transfers to external facilities under the pretence of needing specialised care, using these demands to leave correctional premises temporarily, avoid court appearances, or evade disciplinary actions.

‘Some incarcerated people who are clever enough will take ARV treatment and place it inside their pocket and then throw the empty containers to nurses, saying that they no longer drink ARV medication when they demand to be transferred to another correctional centre.’(44-year-old professional nurse)

‘Okay, some incarcerated individuals living with HIV and AIDS will refuse to take ARVs on an empty stomach, alleging that they are on a hunger strike. The reasons for embarking on a hunger strike are related to demands to be transferred to the Department of Correctional Services Management Area A, or demands to be placed on the Parole Board Assessment Committee List.’(35-year-old professional nurse)

Health becomes instrumentalised and incarcerated individuals’ control beliefs centre on navigating institutions rather than managing health. Their coping strategies prioritise institutional goals (transfer, parole), with medication adherence treated as leverage. This reflects how incarceration distorts the therapeutic purpose of healthcare, transforming it into a transactional system where health status becomes the currency for institutional benefits.

The CSM framework reveals how correctional environments fundamentally distort illness representations for both incarcerated individuals and healthcare professionals, including nurses. For incarcerated individuals, institutional pressures (violence, food insecurity, power dynamics) shape the cognitive and emotional perceptions of HIV/ART. Illness identity becomes secondary to institutional identity; consequence beliefs prioritise immediate survival over long-term health; control beliefs focus on manipulating systems rather than managing disease; and coping strategies become maladaptive, weaponising adherence for institutional gain.

For nurses, systemic barriers (policy gaps, inadequate training, and manipulation experiences) undermine control beliefs and generate emotional responses of exhaustion, frustration, and moral distress. These emotional representations affect their coping strategies, which could shift from patient-centred care to defensive practices. The absence of institutional support leaves nurses unable to develop adaptive coping mechanisms, perpetuating cycles of ineffective adherence management.

These findings demonstrate that adherence challenges in correctional settings are not just individual behavioural issues but systemic problems stemming from institutional structures that fundamentally alter how nurses and incarcerated individuals perceive, experience, and respond to HIV/ART treatment.

#### 3.3.4. Discussion

This study explored the experiences of professional nurses regarding antiretroviral adherence of incarcerated people living with HIV/AIDS in South African correctional facilities. Three themes emerged: challenges in the management of foreign-born incarcerated individuals, navigating manipulative behaviours, and preventing the misuse of ARV medications. These findings reveal how institutional structures shape adherence behaviours for both nurses and incarcerated individuals.

The lack of policies for managing foreign national incarcerated individuals creates fundamental barriers to care continuity. Without documentation verification systems or cross-border communication mechanisms, nurses make ad hoc decisions without institutional support. This policy gap aligns with Young’s [26] findings on structural barriers in correctional health systems and UNAIDS documentation on access barriers facing migrants in institutional settings. The inability to conduct treatment follow-ups across borders compromises individual outcomes and enables the development of drug resistance, creating moral distress for nurses who lack the tools to provide adequate care.

Adherence challenges stem from systemic rather than individual factors [27]. Overcrowding, resource limitations, and inadequate infrastructure hinder the provision of confidential and consistent ART services [28,29]. A lack of private spaces breaches confidentiality and erodes trust, while inconsistencies in medication supply and staffing shortages disrupt treatment continuity. Prison power dynamics, which prioritize security over health [30], restrict nurses’ clinical autonomy, creating ethical ambiguity and moral distress [31,32]. Nurses described feeling isolated and unprepared for correctional healthcare’s psychological complexities [32,33,34,35,36,37,38,39], consistent with literature on high-stress correctional environments with limited professional development.

Incarcerated individuals strategically leveraged non-adherence to negotiate privileges, exploit grievance systems, and avoid consequences. While emotionally taxing for nurses, manipulation represents a learned survival strategy within institutional power structures [8,40] rather than simple non-compliance. When basic needs remain unmet and protection from violence is unavailable, incarcerated individuals instrumentalize health status to gain resources or safety [41,42]. The weaponization of therapeutic diets reflects fundamental food insecurity; medication exchange for protection from sexual violence reveals institutional failure to ensure safety. These behaviors require reframing as adaptive responses to deprivation rather than individual pathology.

ARV diversion for substance use or trading threatens public health through drug resistance and treatment failure. Medication smoking, exchange for drugs or protection, and strategic hoarding reflect disproportionately prevalent substance use disorders in correctional populations [43,44,45,46] and inadequate treatment programs. Medication theft by gangs [47,48,49,50,51], food insecurity [52,53,54,55], and unplanned transfers further compromise adherence, demonstrating how institutional vulnerabilities, violence, deprivation, and poor coordination create conditions where misuse becomes a survival strategy.

As nurse-researchers with experience in correctional healthcare, our insider status facilitated rapport but also risked interpretive bias. Our professional identity may have emphasized structural barriers over individual accountability, while institutional familiarity potentially normalized certain practices without critical examination [56,57,58]. We addressed these influences through reflexive journaling, external co-coding, and systematic comparison with offender-focused literature. Our analytical reframing of “manipulation” as a survival strategy represents an interpretive choice privileging structural over individual explanations. This transparency acknowledges findings as co-constructed knowledge, shaped by participant experiences, institutional contexts, and the researcher’s positionalities [23,56], thereby enhancing credibility through methodological accountability.

This study demonstrates that correctional healthcare is fundamentally shaped by institutional structures, which in turn transform professional practice and patient behavior. Nurses function within dual accountability systems, creating ethical ambiguity [30,57]; incarcerated individuals navigate deprivation and violence, reshaping illness perceptions. Standard adherence frameworks often prove inadequate when healthcare becomes a transactional currency. Effective interventions require systemic transformation addressing policy development, resource allocation, and institutional culture. Individual-level counseling cannot succeed within environments characterized by food insecurity, violence, and policy vacuums. Policy reforms must enhance care quality while upholding dignity, supported by adequate resources, clear protocols, and institutional prioritization of health alongside security [31,32].

### 3.4. Recommendations Were Made to the Department of Health and Policy Frameworks, as Well as to Nursing Practice

Based on study findings revealing systemic barriers to antiretroviral adherence in correctional facilities, the following recommendations are directed to the Department of Health, Department of Correctional Services, nursing practice, and the research community. These recommendations address policy development, clinical practice enhancement, and future research priorities essential for improving ART adherence outcomes among incarcerated individuals living with HIV/AIDS.

Policy Development: Develop standardized protocols for foreign national incarcerated individuals, including cross-border treatment verification and memoranda with neighboring countries. Implement anti-diversion strategies through mandatory DOT for high-risk incarcerated individuals, tamper-evident packaging, and security coordination. Reform food security policies to eliminate the weaponization of therapeutic diets and grievance systems, thereby preventing abuse while protecting rights. Integrate substance abuse programs and ensure adequate staffing ratios, confidential spaces, and uninterrupted medication supplies.

Nursing Practice: Provide specialized training in manipulation recognition, de-escalation, ethical decision-making, and cultural competency. Establish clinical supervision and peer support systems addressing moral distress and burnout. Implement structured adherence protocols that incorporate risk assessment tools, individualized care plans, and post-release linkage services. Strengthen multidisciplinary collaboration and empower nurses through clinical autonomy and grievance protection.

Future Research Priorities

Conduct mixed-methods research evaluating intervention effectiveness, including controlled trials of nurse training programs and anti-diversion strategies. Develop and evaluate evidence-based educational interventions for nurses, incarcerated individuals, and correctional staff. Examine policy implementation barriers, cost-effectiveness, and comparative adherence strategies across settings. Explore offender perspectives through qualitative and participatory action research.

### 3.5. Limitations

The study was conducted during the COVID-19 outbreak, when movements and contacts were restricted; therefore, face-to-face interviews were not feasible due to COVID-19 regulations. Despite these limitations, the study offers a meaningful and in-depth understanding of the challenges and barriers professional nurses face in adhering to ARVs and supporting ARV adherence in correctional services. These insights offer a valuable foundation for future research and for the development of interventions and strategies aimed at improving medication adherence outcomes for incarcerated individuals living with HIV and AIDS in correctional

## 4. Conclusions

This qualitative study explored professional nurses’ experiences regarding antiretroviral adherence by incarcerated individuals living with HIV/AIDS in South African correctional facilities, revealing critical systemic barriers threatening individual health outcomes and broader public health goals. Three major challenges emerged: managing foreign national incarcerated individuals without standardized protocols, navigating manipulative behaviors, and preventing ARV medication misuse, all compromising adherence and increasing risks of drug resistance, treatment failure, and HIV transmission.

This study contributes importantly to correctional healthcare literature by: (1) providing empirical evidence of how institutional structures reshape illness perceptions and coping behaviors through Leventhal’s Common-Sense Model; (2) documenting previously underexplored experiences of South African correctional nurses; and (3) demonstrating how correctional environments transform healthcare into a transactional system where adherence becomes institutional currency rather than health preservation. These insights challenge individual-focused interventions and demand systemic reform.

Achieving optimal ART adherence in correctional facilities requires transforming systemic structures that currently undermine health behaviors. This study provides evidence that individual-level interventions alone cannot succeed within institutional environments characterized by policy gaps, resource constraints, power imbalances, and competing priorities. Effective solutions demand coordinated action across policy development, healthcare delivery, correctional administration, and community partnerships.

The evidence presented here should prompt policymakers, healthcare administrators, and correctional leaders to take immediate action and implement the recommended interventions. Simultaneously, researchers must advance the evidence base through rigorous evaluation of the effectiveness, cost-effectiveness, and sustainability of these interventions. Only through such comprehensive, evidence-based approaches can we protect the health and dignity of incarcerated individuals, support the professional nurses who serve them, and safeguard broader public health, ultimately contributing to the global goal of ending the HIV/AIDS epidemic by 2030.

## Figures and Tables

**Table 1 ijerph-22-01772-t001:** Demographic profile of the participants.

Criterion of Participants	Age	Sex	Marital Status	Level of Education	Religion	Ethnicity
Participants 1	30–39 years	Female	Widowed	Master’s degree	Apostolic Faith Mission	Sepedi
Participants 2	40–49 years	Female	Single	Honours degree	Zion Christian Church	Tsonga
Participants 3	30–39 years	Female	Single	Honours degree	Zion Christian Church	Tsonga
Participants 4	30–39 years	Male	Married	Diploma	Full Gospel	Venda
Participants 5	50–59 years	Female	Widowed	Diploma	Kingdom Life	Sepedi
Participants 6	50–59 years	Female	Married	Master’s degree	Zion Christian Church	Venda
Participants 7	40–49 years	Female	Single	Honours degree	Non-affiliated	Venda
Participants 8	30–39 years	Female	Married	Diploma	Apostolic Faith Mission	Venda
Participants 9	40–49 years	Male	Single	Diploma	Redeeming Showers	Tsonga
Participants 10	50–59 years	Female	Single	Diploma	Christian Fellowship	Venda
Participants 11	30–39 years	Female	Married	Master’s degree	Redeeming Showers	Venda
Participants 12	40–49 years	Female	Married	Diploma	Christian Fellowship	Venda
Participants 13	30–39 years	Female	Married	Honours degree	Redeeming Showers	Sepedi
Participants 14	50–59 years	Female	Married	Honours degree	Apostolic Faith Mission	Venda
Participants 15	30–39 years	Female	Married	Diploma	Apostolic Faith Mission	Venda
Participants 16	50–59 years	Female	Widowed	Diploma	Full Gospel	Venda
Participants 17	40–49 years	Female	Married	Diploma	Kingdom Life	Tsonga
Participants 18	30–39 years	Female	Married	Diploma	Full Gospel	Tsonga
Participants 19	30–39 years	Male	Married	Diploma	Kingdom Life	Sepedi

**Table 2 ijerph-22-01772-t002:** Themes and subthemes emerged from the results (detailed in Appendix A).

Themes	Subthemes
1. PNs’ experiences regarding incarcerated individuals from foreign countries.	1.1. Lack of policies addressing the needs of foreign nationals. 1.2. Staff unpreparedness
2. PNs’ experiences of incarcerated individuals’ manipulative behaviours	2.1. System manipulation for personal gain. 2.2. Exploiting legal procedures. 2.3. Misuse of grievance systems
3. PNs’ experiences of misusing of ARV medication by Incarcerated individuals.	3.1. Faking a medical condition 3.2. Misuse of Prescribed Medication 3.3. Demanding Unnecessary transfers

## Data Availability

Data analysed in this study can be obtained from the first author upon reasonable request.

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
