# Peer review of "Professional Nurses’ Experiences Regarding Antiretroviral Adherence by Incarcerated Individuals Living with HIV and AIDS in Correctional Services"

_ijerph, 2025, doi:10.3390/ijerph22121772_

Round 1
Reviewer 1 Report
Comments and Suggestions for Authors
I would like to begin by congratulating the authors on their work. This is a topic of great interest for professionals who work with this population group, such as inmates. In the field of HIV infection, adherence to antiretroviral treatment is of paramount importance.
Although in other European countries, such as Spain, the prevalence of HIV in prisons does not exceed 3% of the population, it is higher in South Africa. For this reason, the topic is of even greater interest, and studying the opinions of nursing professionals responsible for monitoring treatment adherence is an excellent idea to identify areas for improvement.
Methodologically, the study is well structured. The selection process was appropriately carried out. The sample size (22 nurses) exceeds the number of participants usually required once information saturation is reached (19 nurses).
It does seem somewhat unusual that the pilot study was conducted with only two people. However, if this helped the authors correct any issues with the questionnaire, it can be considered valid.
It would be very helpful if the authors included a simple table listing the different questions used in the semi-structured survey. This would enhance the clarity of the article and make it easier for other researchers to replicate the study.
The discussion and conclusions are appropriate. The only suggestion I would add is to consider conducting further research on this topic in the future.
The bibliography is extensive for an original article and up to date.
In my opinion, this is a study that can be published once the suggested revisions have been considered.
Author Response
Comment 1 Methodologically, the study is well structured. The selection process was appropriately carried out.
Response 1 The sample size (22 nurses) exceeds the number of participants usually required once information saturation is reached (19 nurses). The number of participants corrected, number (19) written on page 3, line 128.
Comment 2 It does seem somewhat unusual that the pilot study was conducted with only two people. However, if this helped the authors correct any issues with the questionnaire, it can be considered valid
Response 2 Four female participants were interviewed for the pilot study from the two correctional service Management Areas, and adjustments were made thereafter, written on page 4, lines 159-161
Comment 3 It would be very helpful if the authors included a simple table listing the different questions used in the semi-structured survey. This would enhance the clarity of the article and make it easier for other researchers to replicate the study.
Response 3 Different probing questions, written on page 4, lines 170-180.
Comment 4 The discussion and conclusions are appropriate. The only suggestion I would add is to consider conducting further research on this topic in the future.
Response 4 The word “future research” and additional information regarding the study's future research on recommendation, written on 15, line, 582 and lines 592-600.
All references updated
Reviewer 2 Report
Comments and Suggestions for Authors
The article “Professional Nurses’ Experiences Regarding Antiretroviral Adherence by Offenders Living with HIV and AIDS in Correctional Services” presents a qualitative study conducted in the Limpopo province, South Africa, with the aim of understanding nurses' experiences regarding adherence to antiretroviral treatment among incarcerated individuals living with HIV/AIDS.
The topic is relevant and current, as it addresses a vulnerable and understudied population, highlighting challenges faced by healthcare professionals in prison settings. The study uses an exploratory and descriptive approach, with semi-structured interviews, which is suitable for capturing perceptions and experiences. The results were organized into three main themes: difficulties related to foreign prisoners, manipulative behaviors of inmates, and misuse of antiretroviral medication. These findings reveal institutional, ethical, and behavioral barriers that hinder adherence to treatment within prisons.
Despite its relevance, the article has some limitations. The theoretical foundation is superficial and lacks conceptual models to support the analysis, such as adherence or health behavior theories. The discussion of the results is more descriptive than critical, with little problematization of the structural and social causes of the reported behaviors. Furthermore, the study lacks reflexivity, as it does not discuss how the position of nurses or researchers may influence the perceptions presented.
From a methodological point of view, the sampling and analysis process are adequately described, but details are lacking regarding data validation and strategies for ensuring scientific rigor, such as triangulation or peer validation (see COREQ guideline). Ethically, the study follows the basic principles of consent and confidentiality, but the discourse on "manipulation" of prisoners could be treated with greater sensitivity and critical reflection.
In terms of contribution, the article gives visibility to the role of nurses in contexts of deprivation of liberty and highlights political and structural gaps in the management of HIV in correctional institutions. However, the recommendations presented — such as strengthening policies and training — are generic and poorly grounded in practical evidence.
In short, this is a relevant work, well-structured and consistent with its objectives, but which presents theoretical and analytical weaknesses. The study contributes empirically to the knowledge about adherence to antiretroviral treatment in prisons, but needs greater critical and conceptual depth to broaden its scientific and social impact.
Author Response
Comment 1 Despite its relevance, the article has some limitations. The theoretical foundation is superficial and lacks conceptual models to support the analysis, such as adherence or health behavior theories.
Response 1 The theoretical foundation, written on pages 2-3, lines 85-101.
Comment 2 From a methodological point of view, the sampling and analysis process are adequately described, but details are lacking regarding data validation and strategies for ensuring scientific rigor, such as triangulation or peer validation (see COREQ guideline).
Response 2 Details of data validation and strategies for ensuring scientific rigor, as well as the COREQ guideline. Written on page 5, lines 206-236.
Comment 3 The discussion of the results is more descriptive than critical, with little problematization of the structural and social causes of the reported behaviors. Furthermore, the study lacks reflexivity, as it does not discuss how the position of nurses or researchers may influence the perceptions presented.
Response 3 Table 1.1 restructured on page 6, line 239. Information regarding the structural and social causes of the reported behaviors, written on page 11, lines 457-475
Information regarding the position of authors influences the perceptions presented, written on page 5, lines 230-240
Comment 4 However, the recommendations presented — such as strengthening policies and training are generic and poorly grounded in practical evidence. In short, this is a relevant work, well-structured and consistent with its objectives, but it presents theoretical and analytical weaknesses.
Response 4 Additional information regarding strengthening policies and training, written on page 14, lines 592-600
Comment 5 The study contributes empirically to the knowledge about adherence to antiretroviral treatment in prisons, but needs greater critical and conceptual depth to broaden its scientific and social impact.
Response 5 Information regarding critical and conceptual depth to broaden its scientific and social impact. Written on page 14, lines 565-581
All references updated
Round 2
Reviewer 2 Report
Comments and Suggestions for Authors
The article presents an exploratory-descriptive qualitative study that seeks to understand the experiences of professional nurses regarding adherence to antiretroviral therapy (ART) among people deprived of liberty living with HIV/AIDS in correctional units in Limpopo Province, South Africa.
The article's title is clear and informative, directly presenting the theme, the phenomenon, and the target audience of the study. However, it is excessively long and could be simplified without loss of meaning, which would contribute to greater objectivity and impact. The abstract, in turn, partially meets the journal's requirements. It is structured in typical sections—introduction, method, results, and conclusion—but is still excessively descriptive. It lacks relevant quantitative and contextual information, such as the exact number of participants, the form of data analysis, and the theoretical basis that supports the study. In addition, the abstract does not highlight in a sufficiently clear way the practical implications of the findings for public policy and nursing practice, something valued by IJERPH, which prioritizes studies applicable to the promotion of public health.
The article's introduction satisfactorily fulfills its contextualizing function. The text presents a global and local overview of HIV/AIDS, highlighting the impact of the epidemic in Sub-Saharan Africa and, particularly, in the South African prison system. This contextualization is well supported by up-to-date and relevant references, most of which were published between 2020 and 2024, demonstrating attention to current scientific trends. The research problem is identified and justified, but the study's objective could be formulated more directly and succinctly, in a single sentence that clearly states the investigative purpose. The theoretical framework, based on Leventhal's Common Sense Model, is a strong point of the article, as it provides a coherent conceptual structure for analyzing nurses' perceptions and experiences. However, the introduction is excessively long, repeats information, and could be condensed to make the reading more fluid and objective.
The methodology section is one of the most robust in the article and demonstrates care in meeting the quality criteria for qualitative research recommended by IJERPH and COREQ. The study design is consistent with the proposed objectives, and the description of the procedures is detailed. The article clearly states the type of study, the methodological approach, the context, the inclusion and exclusion criteria, and the purposive sampling process. The final sample of nineteen participants is adequate, considering the data saturation, although the text presents a minor inconsistency: it mentions twenty-two interviews at one point, which requires clarification.
The data collection process, carried out through semi-structured telephone interviews, is well described and justified, especially since it occurred in a context of health restrictions resulting from the COVID-19 pandemic. Data analysis followed Tesch's method, appropriate for qualitative studies, but the article could specify more precisely whether any analytical support software was used and how the codes and themes were validated among the researchers. Ethical aspects are extensively detailed, with information on ethics committee approval, informed consent, and confidentiality, fully meeting editorial requirements.
The results are organized into three major themes and their respective subthemes, which facilitates reader comprehension. The participants' statements are transcribed faithfully and adequately illustrate the identified categories. However, the section is excessively long and becomes repetitive in certain parts. The description is rich, but it lacks a more interpretive and integrative analysis of the findings, articulating the reported experiences with the theoretical model that guided the study. In other words, the results section favors the exposition of the accounts at the expense of analytical reflection. The table with the themes and subthemes is useful, but it could be improved with the inclusion of examples of citations and interpreted meanings, which would contribute to a more synthetic reading and to the demonstration of the rigor of the analysis.
The discussion is one of the most consistent parts of the article, demonstrating a good articulation between the results obtained and recent scientific literature. The text demonstrates theoretical and contextual mastery, relating the findings to the structural difficulties of the South African prison system, the working conditions of nurses, and the psychosocial factors that influence adherence to ART. There is a clear effort at critical interpretation and conceptual integration, in line with the journal's guidelines. However, the discussion could be more concise and less repetitive, as some arguments are revisited almost identically throughout the text. It would also be desirable to delve deeper into the analysis of the researchers' reflexivity—an important aspect in qualitative research—especially with regard to the possible influence of their professional and institutional position on the collection and interpretation of data.
The conclusion adequately summarizes the main findings and reinforces the relevance of the study for the formulation of health policies in the prison context. However, a more assertive and proactive conclusion is still lacking, one that clearly highlights the contributions of the work and its practical implications for nursing and public health. Furthermore, the manuscript itself contains an editorial note requesting the inclusion of a paragraph on future research directions, which reinforces the need to explicitly suggest concrete approaches to scientific continuity. The study, due to its qualitative nature, could indicate avenues for mixed-methods research or educational interventions aimed at nurses and prison managers, in order to align with the applied and socially impactful profile that IJERPH seeks in its articles.
In terms of form and style, the manuscript mainly follows the structure required by the journal (Abstract, Introduction, Methods, Results, Discussion, Conclusion, References). However, there are inconsistencies in the standardization of references, especially regarding the standardization of authors, years, and the inclusion of DOIs. The text presents minor linguistic flaws and repetitions that require language revision, preferably by a native speaker of academic English. These formal details, although minor, compromise the professional presentation of the manuscript.
In summary, the article presents unquestionable scientific merit and social relevance, addressing a sensitive and little-explored topic, combining methodological rigor and practical pertinence. This study contributes to the understanding of the complex dynamics between care, therapeutic adherence, and the prison context, offering important insights for improving public health policies and professional nursing training.
Author Response
Writefull was used to revise the text and to improve language corrections as indicated in the comments
TABLE FOR ADDRESSING REVIEWER 2’s COMMENTS
|
ITEM |
REVIEWERS’ COMMENT |
AUTHORS’ RESPONSES |
|
1 |
In addition, the abstract does not highlight in a sufficiently clear way the practical implications of the findings for public policy and nursing practice, something valued by IJERPH, which prioritizes studies applicable to the promotion of public health. |
The abstract has been revised, and the missing section has been added in lines 21-30 |
|
2 |
The theoretical framework, based on Leventhal's Common Sense Model, is a strong point of the article, as it provides a coherent conceptual structure for analyzing nurses' perceptions and experiences. However, the introduction is excessively long, repeats information, and could be condensed to make the reading more fluid and objective. |
The introduction has been revised and shortened in lines 35-64 |
|
3 |
The final sample of nineteen participants is adequate, considering the data saturation, although the text presents a minor inconsistency: it mentions twenty-two interviews at one point, which requires clarification. |
Addressed in line 90 |
|
4 |
Data analysis followed Tesch's method, appropriate for qualitative studies, but the article could specify more precisely whether any analytical support software was used and how the codes and themes were validated among the researchers. |
Information that no software analysis was used has been added in line 154. |
|
6 |
The results are organized into three major themes and their respective subthemes, which facilitates reader comprehension. The participants' statements are transcribed faithfully and adequately illustrate the identified categories. However, the section is excessively long and becomes repetitive in certain parts. The description is rich, but it lacks a more interpretive and integrative analysis of the findings, articulating the reported experiences with the theoretical model that guided the study. In other words, the results section favors the exposition of the accounts at the expense of analytical reflection. The table with the themes and subthemes is useful, but it could be improved with the inclusion of examples of citations and interpreted meanings, which would contribute to a more synthetic reading and to the demonstration of the rigor of the analysis.
|
The table has been enhanced The results section has been revised in lines 215-218, 223-229, The results have been improved to integrate the CSM at the end of each theme, from lines 245-251, 269-274,295-301, 323-328,360-361, 370-375,412-433,
|
|
|
However, the discussion could be more concise and less repetitive, as some arguments are revisited almost identically throughout the text. It would also be desirable to delve deeper into the analysis of the researchers' reflexivity—an important aspect in qualitative research—especially with regard to the possible influence of their professional and institutional position on the collection and interpretation of data. |
The discussion is revised to be concise with deeper analysis and reflexivity added in lines 435-495 |
|
|
The conclusion adequately summarizes the main findings and reinforces the relevance of the study for the formulation of health policies in the prison context. However, a more assertive and proactive conclusion is still lacking, one that clearly highlights the contributions of the work and its practical implications for nursing and public health. Furthermore, the manuscript itself contains an editorial note requesting the inclusion of a paragraph on future research directions, which reinforces the need to explicitly suggest concrete approaches to scientific continuity. The study, due to its qualitative nature, could indicate avenues for mixed-methods research or educational interventions aimed at nurses and prison managers, in order to align with the applied and socially impactful profile that IJERPH seeks in its articles |
Recommendations has of future research have been added in lines 504-523
The conclusion has been improved to expose assertiveness from lines 548-523 |
